

# Brief communication: Rethinking the 1998 China flood to prepare for a nonstationary future

Shiqiang Du[1,2], Xiaotao Cheng[3], Qingxu Huang[4], Ruishan Chen[5], Philip Ward[2], and Jeroen C. J. H. Aerts[2]

[1]School of Environmental and Geographical Sciences, Shanghai Normal University, Shanghai, 200234, China
[2]Institute for Environmental Studies, Vrije Universiteit Amsterdam, Amsterdam, 1081 HV, The Netherlands
[3]China Institute of Water Resources and Hydropower Research, Beijing, 100038, China
[4]State Key Laboratory of Earth Surface Processes & Resource Ecology, Beijing Normal University, Beijing, 100875, China
[5]School of Geographic Sciences, East China Normal University, Shanghai, 200241, China

*Correspondence to*: Shiqiang Du (shiqiangdu@shnu.edu.cn); Jeroen C. J. H. Aerts (jeroen.aerts@vu.nl)

**Abstract.** A mega-flood in 1998 caused tremendous losses in China, and triggered large investments in flood risk management (Bryan et al., 2018). However, rapid urbanization and climate change pose new challenges and it is time to rethink whether China is prepared for the next mega-flood. In China's fast growing economy, with rapid urbanization, novel flood risk management approaches are required in addition to reinforcing structural protection, such as levees. These include a risk-based urban planning and a coordinated water governance system with public participation.

## 1. China's mega-flood in 1998

In 1998, a mega-flood swept through China's major river basins, including the Yangtze, Songhua, Nei, Min, and Pearl Rivers. In the Yangtze and Songhua Rivers, floodwaters exceeded historical maximum heights and overtopped 300 km of dikes, and about 15,000 dike segments were in an emergency state, having to be reinforced by 8 million rescuers. Nationwide, the floods affected 186 million people, caused 4,150 deaths, and led to total economic losses of US$ 70 billion (in 2015 US$) (Kobayashi
and Porter, 2012). Whilst the mega-flood of 1998 had particularly severe impacts, the entire 1990s also saw large losses. On average, economic damages from floods in the decade were around US$ 40 billion per year, accounting for 2.28% of China's GDP.

The main drivers of the disastrous 1998 flood are considered to be land use change and bad maintenance of levees, apart from
the extreme weather conditions (Nakayama and Watanabe, 2008). China extensively exploits its land to feed 21% of the world's population, having only 6% of the world's total water resources and 9% of the world's arable land. This has resulted in a rapid degradation of the forested upper catchments disrupting the functioning of reservoirs through enhanced peak flows and increased soil erosion. In the middle and lower reaches, wetlands and waterbodies were reclaimed as polders for farmlands or fishponds. As a result, the capacity of wetlands and waterbodies to store and to convey floodwaters declined. To protect
polders against rising floodwaters, lakes and tributaries were further disconnected from river channels by flood gates. For





example, the lake surface area connected to the Yangtze River reduced from 17,198 km$^2$ in the 1950s to 6,000 km$^2$ in 1998 —
this increased flood water levels in 1998 by approximately one meter (Ministry of Water Resources, 1999).

## 2. China's response to the 1998 flood

In response to the mega-flood, China adopted a series of integrated flood management policies, focusing on three major issues:
5 1) conserving soil and water through forest protection and reforestation; 2) returning reclaimed lands to open water and
wetlands, and; 3) enhancing both levee and reservoir systems to increase flood protection and control. In the years after the
1998 flood, 16 major sustainability programs (Bryan et al., 2018) were launched. For instance, the Grain for Green Program
(1999–2020) aims to prevent soil erosion and mitigate flooding by converting cropland and wasteland on hillslopes into natural
forests and grasslands. The programs related to runoff and erosion invested a total of 114.2 billion US$ from 1998–2015,
10 accounting for 32.5% of the total sustainability program investment (Bryan et al., 2018). These efforts reduced soil erosion
nationwide with 12.9% between 2000 to 2010, and with 58.8% and 27.0% in the Yangtze and Yellow river basins, respectively
(Deng et al., 2012). Furthermore, the capacity of wetlands to temporarily store flood waters improved by 12.7% (Ouyang et
al., 2016).

15 China also required local communities to convert polders into wetland areas and lakes for capturing floodwaters. From 1998–
2002, 1,461 polders were removed and 2.4 million people were relocated elsewhere in the Yangtze River basin. This increased
the inland water area by 2,900 km$^2$ and added a storage capacity of 13 billion m$^3$ (Ministry of Water Resources, 2015). However,
efforts to restore and protect open water, such as lakes and ponds, deteriorated over time (Cheng and Li, 2015), and by 2015
there are still 406 polders in the main channels of Yangtze River and 133 polders in the Dongting- and Poyang lake regions,
20 with a total population of 1.9 million (Ministry of Water Resources, 2015). Nationwide, the population living in floodplains
increased by 1.1% per year over the period 1990–2015, which is much faster than the population increase outside floodplains
(0.4% per year) (Fang et al., 2018).

The impact of the 1998 floods also led to a new flood protection program, and China invested a total of US$ 294 billion on
25 technical protection during 1998–2017, accounting for more than a third of the total investment on water engineering that also
includes water supply and hydropower generation (Fig. 1). As a result, the lengths of the river banks protected by up-to-
standard dikes rose from 76,532 km in 1998 to 201,124 km in 2016, and the reservoir capacity increased from 493 trillion to
897 trillion m$^3$ (Ministry of Water Resources, 2017). For example, the well-known Three Gorges Dam, which was completed
in 2006, has a capacity of 39.3 billion m$^3$ and increases the protection standard from a 10-year- to a 100-year flood at Jingzhou,
30 a weak point in the Yangtze river downstream from the dam (Mei, 2010).





Finally, China has also made huge efforts to improve evacuation, and in the 2010s, China annually evacuated 9.9 million people and called in 10.8 million rescuers for emergencies. For example, 190,000 people were evacuated in Shanghai for Typhoon Anbi on July 22th, 2018.

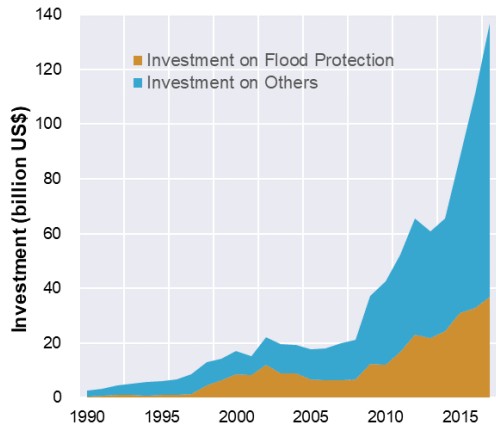

**Figure 1:** Investment in flood protection and other water engineering programs (e.g. water supply and hydropower) in China

## 3. Societal effects of response measures after 1998

All flood management efforts after the 1998 flood initially decreased the annual flood fatalities from 3909 in the 1990s to 1454 in the 2000s, but the number levelled at 1134 in the 2010s (Fig 2 a). Furthermore, medium- and small-size basins are disproportionally suffering flood fatalities, with 15,662 deaths in the 2010s (72.6% of the total flood fatalities in China). For instance, a local-scale catastrophe in Zhouqu of western China led to 1765 fatalities on August 8th, 2010, marking 2010 as the worst year since 1998 in terms of flood fatalities.

Furthermore, despite the flood management investments, flood damages reached a new peak during the 2010s, with US$ 41.1 billion per year, exceeding the long term annual averages of US$ 24–39 billion in the 1990–2000s (Fig 2b). While the ratio of economic losses from flooding to GDP dropped from 2.3% in the 1990s to 0.6% in the 2000s, and to 0.4% in the 2010s, the current loss ratio (0.4%) is much larger than in developed countries (e.g. <0.05% in the USA (Roger Pielke, 2015)). When focusing on flood risk in Chinese cities, floods have annually hit about 157 cities since 2006, and this number is increasing. Flood events in highly urbanized areas have caused large indirect economic ripple effects, and there are recent examples of paralyzed cities in China due to flood impacts to critical infrastructure. In recent years, this has annually interrupted ~40,000 factories, affected electricity supply ~20,000 times, and shut down 166 airports and seaports (Ministry of Water Resources, 2017).



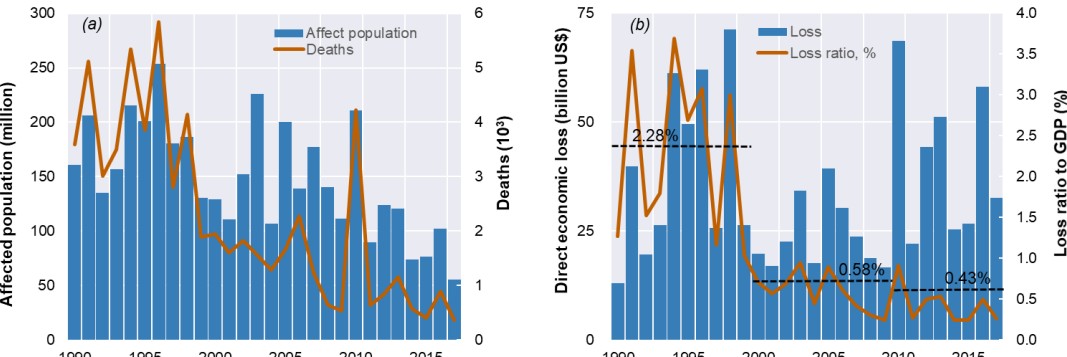

**Figure 2:** Flood fatalities (a) and flood losses (b) in China from 1990–2017

## 4. Future challenges to flood management in China

It is expected that future urbanization and climate change will further increase economic flood damage by 263%−331% in the Yangtze River in the year 2080 (Winsemius et al., 2016), as compared to 2010 levels. For China as a whole, it is expected that economic production losses from floods will increase more rapidly than other countries (Willner et al., 2018). If the world's second largest economy continues suffering huge flood damages, the impacts could be felt worldwide and may hamper the global economy. During 2016–2035, China is expected to suffer two-thirds of the global direct production losses caused by floods, US$389 billion, which can further propagate an indirect impact of about US$300 billion to other countries through the global trade and supply network.

Flood management in China, and in particular in urban areas, could be better prepared for these trends. Urbanization in Chinese floodplains continues to increase, for example at rate of 20,600 km$^2$/year over the period 1998 and 2015 (Du et al., 2018), and by 2050, it is expected that 80.0% of China's total population will live in cities, rising from 55.5% in 2015 (Population Division of United Nations, 2018) — with an additional of 172.8 million people in floodplains (Fang et al., 2018). These numbers will probably be even higher due to the ending of the 'one-child' policy.

Another challenge is to keep up flood protection standards with the pace of urbanization and climate change. A survey in 2013 (Cheng and Li, 2015) showed that 50% (or 321) of the 642 Chinese cities did not reach the required flood protection standards, and 44% (or 284) of the cities did not complete or update flood management plans. This especially holds for cities in the semi-arid northwestern part of China, which have a slower economic development, and generally have limited flood protection. However, many of those cities are rapidly growing in the floodplains (Du et al., 2018; Fang et al., 2018), and are expected to experience increasing extreme precipitation events (Zhou et al., 2014). The floods in Lanzhou, Ningxia, and Xining in 2017 and 2018 are wakeup calls about rising flood risk in this area.





## 5. Future adaptation

Novel flood adaptation policies are required to anticipate these (uncertain-) future challenges. Such policies should be based on a well-established and up to date risk assessment, which account for future changes in climate and socioeconomic conditions. On component of the new policies could be flood protection systems, especially in urban areas with high economic values and large exposed populations (Ward et al., 2017). However, structural measures can also release the 'levee effect', further stimulating exposure in protected areas. Hence, additional measures and regulations are required to solve this paradox between urban development and structural protection, to sustain and enhance environmental values, and to reduce flood risk in areas where dikes are not cost-effective.

*Integrate flood management into urban planning:* The 1998 floods once triggered a large scale of returning farmlands and settlements to wetlands and inland water areas, unprecedented in this populous country. However, the awareness gained in the aftermath of the 1998 catastrophe seems to be dissipating over time as cities again reclaim lands from natural floodplains and urban exposure is again rising. New spatial planning policies, such as zoning and building codes, could be developed and reinforced to reduce local flood risk, involving households and communities. In addition, the importance of ecosystems and 'nature based solutions' should be increasingly acknowledged in reducing flood risk. For this latter aspect, China has nominated 1,060,000 'river chiefs' by June 2018 to protect the natural processes of storing and discharging flood waters, from expanding cities (Smith, 2018). Balancing the huge development pressure and restoring spaces for floodwaters is a challenge, but is critical for integrating flood management into urban planning (Opperman et al., 2009).

*Strengthen governance and coordination:* Another challenge is to improve the development of integrated flood risk management plans and designs, preferably involving all responsible stakeholders (Aerts et al., 2014; Rijke et al., 2012). However, flood risk management in China is still characterized by a top-down administration that is divided in different governmental organizations (Kobayashi and Porter, 2012). At national level, for instance, dike design and maintenance is the responsibility of the Ministry of Water Resources (MWR), wetland protection is managed by the Ministry of Ecology and Environment (MEE), the land use master plan falls within the realm of the Ministry of Land and Resources (MLR), and the urban oriented 'Sponge City' program is coordinated by the Ministry of Housing and Urban-Rural Development (MHURD). In this multi-jurisdiction setting, a new governance structure is needed to effectively coordinate water-soil conservation, wetland protection, dike design, and urban planning for reducing flood risk (Cheng and Li, 2015). Such a new governance system should address decentralized governance approaches, involving the heterogeneities of China's communities. Furthermore, attention should be paid to resolve upstream and downstream conflicts, and differences in protection levels between urban- and rural areas (Han and Kasperson, 2011).



*Improve information sharing and public participation:* Flood risk information for the public is currently scarce in China. Flood hazard maps were produced during 2011–2016, and China plans to invest in high resolution flood information, such as from remote sensing. Following the recommendations of the UNISDR Sendai framework (Mysiak et al., 2016), flood risk information sharing facilitates public participation and stimulates a "bottom-up" process to raise awareness and promote local

action for flood adaptation (Haer et al., 2016). In addition, the information on the cost and benefit of wet/dry flood-proofing buildings and other flood adaptation measures must be improved, to demonstrate what local stakeholders can do themselves in terms of flood adaptation. Public participation can also support policy makers developing adaptation measures that have support from the public.

*Acknowledgements.* This research was funded by the National Natural Science Foundation of China (Grant Nos. 41871200, 41730642, 51761135024) and the National Key Research and Development Program of China (2017YFC1503001). We are grateful to Prof. Anders Levermann at Potsdam Institute for Climate Impact Research, Germany for providing the propagation effect of China's production loss to the rest world.

*Author contribution.* Shiqiang Du designed this study and collected the needed information. Shiqiang Du and Jeroen C. J. H. Aerts prepared the manuscript with contributions from all co-authors.

*Competing interests.* The authors declare that they have no conflict of interest.

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
