# Peer review of "Brief communication: Rethinking the 1998 China flood to prepare for a nonstationary future"

_Natural Hazards and Earth System Sciences, 2018_

## Referee Comment (RC1) · Anonymous Referee #1 · 25 Jan 2019

This manuscript presents the brief communication of the 1998 flood in China. This topic is related to the scope of this journal. The authors discussed the rapid urbanization and climate change pose new challenges and rethink whether China is prepared for the next mega-flood. However, this manuscript still need to addresses and clarifies several points before it can be accepted. The following comments may help enhancing the quality of this work. 1. Scientific writing: The manuscript must be professionally proofread and edited. In addition, the authors may pay attention to some aspect of the conventional research writing. Although it is a brief communication, the structure of the manuscript should be enough, especially the connection between the sentences, the components/structure of the key parts (Abstract, Introduction, body, Conclusion). 2. This brief communication discussed the 1998 flood in China, how about the recent

flood, such as flood in 2018 in China. The southern China suffered server floods in 2018. It is suggested to make a comparison between the 1998 flood and 2018 flood in China. The flowing references may be help to strengthen this study. "Flooding hazards across southern China and prospective sustainability measures." 3. The short communication should discussion the persective of flood disaster management, e.g. flood risk assessment and prediction. The following publications are refered: "Flood risk assessment in metro systems of mega-cities using a GIS-based modeling approach" "Assessment of geohazards and preventative countermeasures using AHP incorporated with GIS in Lanzhou" 4. The abstract is too short. Although this is a short communication, the problem, method and results should be included in the abstract.

---

## Referee Comment (RC2) · Anonymous Referee #2 · 20 Feb 2019

Manuscript ID: NHESS-2018-394 Title: Brief communication: Rethinking the 1998 China flood to prepare for a nonstationary future Authors: Shiqiang Du, Xiaotao Cheng, Qingxu Huang, Ruishan Chen, Philip Ward, and Jeroen C. J. H. Aerts

Reviewer's Comment

General comments: This paper reviewed a mega-flood in 1998 which caused tremendous losses in mainland China. Since rapid urbanisation and extreme climate result in great challenges, novel flood risk management is in urgent need. The findings of this study seem to have a guiding role for efficient flood risk management, but there're some issues need to be addressed prior to the acceptance of paper publication in NHESS.

Additionally, the authors may pay attention to some aspect of the conventional research

writing, especially the connection between the sentences, the components/structure of the key parts (Abstract, Introduction, Body, and Conclusions). I suggest the authors may read the following references to modify the paper accordingly. Glasman-Deal, H. (2010). Science Research Writing for non-native speakers of English. Imperial College Press, London, 228p.

Detailed comments: 1) 1 China's mega-flood in 1998: The objectives of this study should be inserted into an appropriate place. This may significantly enhance the readability of this paper.

2) Fig. 1: The authors present the variations in the flood protection and others' investments against the time. However, the data source has not clearly reported yet, which causes a difficulty in convincing general readers to conduct further analysis and/or comparison by retrieving the data presented. Please clarify.

3) Fig. 2: The data source again has not reported yet. Please provide where the data come from and indicate whether the presented data are retrieved from other research.

4) P4, L7-9: The authors indicated that during 2016-2035, China is expected to suffer two-thirds of the global direct production losses caused by floods, US$389 billion, with an indirect impact of about US$300 billion to other countries. No data source available.

5) 5 Future adaptation: The presence of the new challenges forces the development of countermeasures. The authors also list their suggestions against mage flood. Notwithstanding that, details in regard with the mega flood hazard prevention and mitigation are missing. Please elaborate with the details necessary.

6) 6 References: State-of-art researches should be cited and by comparing with the state-of-art researches, the significance of this study should be highlighted. The following research articles would help to make the manuscript more professional and sound; 1. Lyu, H.M., et al. (2018). Flooding hazards across Southern China and perspective sustainability measures. Sustainability-Basel, doi: 10.3390/su10051682. 2. Wang,

Z.F., et al. (2018). Investigation into geohazards during urbanization process of Xi'an, China. Natural Hazards, doi:10.1007/s11069-018-3280-5.
* * *

---

## Author Comment (AC1) · 18 Mar 2019

**Author's Responses to Comments from the Anonymous Referee 1**

Comments made by Anonymous Referee 1 are provided in black text.
Author responses are provided in blue text (line and page numbers refer to the clean version).

***General Comments.*** This manuscript presents the brief communication of the 1998 flood in China. This topic is related to the scope of this journal. The authors discussed the rapid urbanization and climate change pose new challenges and rethink whether China is prepared for the next mega-flood. However, this manuscript still need to addresses and clarifies several points before it can be accepted. The following comments may help enhancing the quality of this work.

*Accepted*: Thanks for confirming the relevance of our manuscript and the suggestions for further improvement. We have thoroughly revised our paper, addressing your valuable comments and suggestions.

***Detailed Comment 1***. Scientific writing: The manuscript must be professionally proofread and edited. In addition, the authors may pay attention to some aspect of the conventional research writing. Although it is a brief communication, the structure of the manuscript should be enough, especially the connection between the sentences, the components/structure of the key parts (Abstract, Introduction, body, Conclusion).

*Accepted*: Thanks for the suggestion. After revising the manuscript, an additional, thorough, proofreading of the manuscript has been carried out by one of the co-authors (Ward), who is a native English speaker and has extensive publication experience in scientific journals. Furthermore, in the revised version we improved the following aspects:

- We rewrote the Abstract to make it more self-explanatory (also following your *Comment 4*).
- We have improved the connections between sentences and sections to enhance the logical flow. With regards the structure, we checked the journal's guidelines and several recent papers of the type 'Brief Communication', and found that the structure is in accordance with these. This structure is typical for an opinion paper with a 'Brief Communication' type; several similar examples are listed below (Please check the references below).

Reference:

Aerts, J. C. J. H.; Botzen, W. J. W., Brief communication "Hurricane Irene: a wake-up call for New York City?". Nat. Hazards Earth Syst. Sci. 2012, 12 (6), 1837-1840.

Mysiak, J., Surminski, S., Thieken, A., Mechler, R., and Aerts, J.: Brief communication: Sendai framework for disaster risk reduction – success or warning sign for Paris?, Nat. Hazards Earth Syst. Sci., 16, 2189-2193, 2016.

Mysiak, J.; Castellari, S.; Kurnik, B.; Swart, R.; Pringle, P.; Schwarze, R.; Wolters, H.; Jeuken, A.; Linden, P. v. d., Brief communication: Strengthening coherence between climate change adaptation and disaster risk reduction. Nat Hazard Earth Sys 2018, 18 (11), 3137-3143.

***Detailed Comment 2***. This brief communication discussed the 1998 flood in China, how about the recent flood, such as flood in 2018 in China. The southern China suffered server

floods in 2018. It is suggested to make a comparison between the 1998 flood and 2018 flood in China. The flowing references may be help to strengthen this study. "Flooding hazards across southern China and prospective sustainability measures."

*Accepted*:    Thanks for this suggestion. We have added references to the floods that occurred in 2018 and in other recent years in the revised manuscript. First, we now discuss how the severe floods in 2018, particularly in western China, were an alarm for the arid/semi-arid region to increase risk awareness and improve adaptation (lines 22–23 on page 4). Second, the revised manuscript includes a new reference to the massive evacuation in Shanghai for Typhoon Anbi on July 22th, 2018 as an example of the huge effects that China has made to improve evacuation (lines 2–3 on page 3). Third, we include new information on flood protection investments from 1998 to 2016 at lines 26–30 on page 2 and in Figure 1 (lines 5–7 on page 3). We also included information on the changes in flood losses between the 1990s and 2010s at lines 9–17 on page 3 and in Figure 2 (lines 1–2 on page 4). Some of the details of the 2018 flood are not included in the figures and the comparison, because detailed data for flood protection investments and flood losses in 2018 have not yet been released. Please note the 2018 flood loses were lower than the past five-year average (The Central Government of PR China, 2019).

Additionally, we added the insightful reference you recommended as further evidence of increasing flood risk in China (lines 15–16 on page 3) and the strategies China should adopt to address the emerging challenges (lines 11–16 on page 5).

Reference:

The Central Government of PR China. 2019. The 2018 natural disaster losses are lower than the average of the past five years. http://www.gov.cn/xinwen/2019-01/10/content_5356406.htm (accessed on February 24, 2018)

*Detailed Comment 3*.    The short communication should discussion the persective of flood disaster management, e.g. flood risk assessment and prediction. The following publications are refered: "Flood risk assessment in metro systems of mega-cities using a GIS-based modeling approach" "Assessment of geohazards and preventative countermeasures using AHP incorporated with GIS in Lanzhou"

*Accepted*:    We enhanced the discussion on flood management with the information of the recommended papers, and added one of them as reference. Note the limited number of references (up to 20) allowed in 'Brief Communications'.

*Detailed Comment 4*.    The abstract is too short. Although this is a short communication, the problem, method and results should be included in the abstract.

*Accepted*:    Thanks for this comment. Following your suggestion, we rewrote the Abstract. Note that there is a 100-word maximum limit, which explains the brevity. It now has 100 words and reads as follows:

> "A mega-flood in 1998 caused tremendous losses in China and triggered major policy adjustments in flood-risk management. This paper aims to retrospectively examine these policy adjustments and discuss how China should adapt to newly emerging flood challenges. We show that China suffers annually from floods,

despite large-scale investments and policy adjustments. Rapid urbanization and climate change will exacerbate future flood risk in China, with cascading impacts on other countries through global trade networks. Therefore, novel flood-risk management approaches are required, such as a risk-based urban planning and coordinated water governance systems with public participation, in addition to traditional structural protection".

---

## Author Comment (AC2) · 18 Mar 2019

**Author's Responses to Comments from the Anonymous Referee 2**

Comments made by Anonymous Referee 2 are shown in black text.
Author responses are provided in blue text (line and page numbers refer to the clean version).

***General comment***:   This paper reviewed a mega-flood in 1998 which caused tremendous losses in mainland China. Since rapid urbanisation and extreme climate result in great challenges, novel flood risk management is in urgent need. The findings of this study seem to have a guiding role for efficient flood risk management, but there're some issues need to be addressed prior to the acceptance of paper publication in NHESS. Additionally, the authors may pay attention to some aspect of the conventional research writing, especially the connection between the sentences, the components/structure of the key parts (Abstract, Introduction, Body, and Conclusions). I suggest the authors may read the following references to modify the paper accordingly. Glasman-Deal, H. (2010). Science Research Writing for non-native speakers of English. Imperial College Press, London, 228p.

*Accepted*:   Thanks for the acknowledgement of the importance of our manuscript and the suggestions for further improvement. We made a thorough revision, performed additional proof reading, and in particular improved the connections between sentences and sections to enhance the logical flow. Note that the chosen structure is typical of type 'Brief Communication' type papers in NHESS; several similar examples are listed below, which are also used in other NHESS papers (Please check the references below).

Reference:
Aerts, J. C. J. H.; Botzen, W. J. W., Brief communication "Hurricane Irene: a wake-up call for New York City?". Nat. Hazards Earth Syst. Sci. 2012, 12 (6), 1837-1840.
Mysiak, J., Surminski, S., Thieken, A., Mechler, R., and Aerts, J.: Brief communication: Sendai framework for disaster risk reduction – success or warning sign for Paris?, Nat. Hazards Earth Syst. Sci., 16, 2189-2193, 2016.
Mysiak, J.; Castellari, S.; Kurnik, B.; Swart, R.; Pringle, P.; Schwarze, R.; Wolters, H.; Jeuken, A.; Linden, P. v. d., Brief communication: Strengthening coherence between climate change adaptation and disaster risk reduction. Nat Hazard Earth Sys 2018, 18 (11), 3137-3143.

***Detailed comment 1.***   China's mega-flood in 1998: The objectives of this study should be inserted into an appropriate place. This may significantly enhance the readability of this paper.

*Accepted*:   Thanks for the good suggestion; we have included the objective more clearly in the revised Abstract and in the manuscript at lines 12–13 on page 1 and lines 5–6 on page 2.

***Detailed comment 2.***   Fig. 1: The authors present the variations in the flood protection and others' investments against the time. However, the data source has not clearly reported yet, which causes a difficulty in convincing general readers to conduct further analysis and/or comparison by retrieving the data presented. Please clarify.

*Accepted*:   The data source is: Ministry of Water Resources: China Water Statistical Yearbook 2017, China Water Power Press, Beijing, 2017. The reference has been added in the revised version (lines 6–7 on page 3).

***Detailed comment 3.***    Fig. 2: The data source again has not reported yet. Please provide where the data come from and indicate whether the presented data are retrieved from other research.

*Accepted*:    The data source is: Ministry of Water Resources: China Water Statistical Yearbook 2017, China Water Power Press, Beijing, 2017. The reference has been added (lines 1 on page 4).

***Detailed comment 4.***    P4, L7-9: The authors indicated that during 2016-2035, China is expected to suffer two-thirds of the global direct production losses caused by floods, US\$389 billion, with an indirect impact of about US\$300 billion to other countries. No data source available.

*Accepted*:    The data source is: Willner, S. N., Otto, C., and Levermann, A.: Global economic response to river floods, Nature Climate Change, 8, 594-598, 2018. The reference has been added (line 9 on page 4).

***Detailed comment 5.***    Future adaptation: The presence of the new challenges forces the development of countermeasures. The authors also list their suggestions against mage flood. Notwithstanding that, details in regard with the mega flood hazard prevention and mitigation are missing. Please elaborate with the details necessary.

*Accepted*:    Thank you for the good suggestion. In the revised version, we have added the following sentences with regards suggestions for flood hazard prevention and mitigation (lines 4–6 on page 5):

> "One component of the new policies could be enhanced flood protection systems, especially in urban areas with high economic values and large exposed populations (Ward et al., 2017). However, structural measures can also release the 'levee effect', further stimulating exposure in protected areas".

***Detailed comment 6.***    References: State-of-art researches should be cited and by comparing with the state-of-art researches, the significance of this study should be highlighted. The following research articles would help to make the manuscript more professional and sound;

1. Lyu, H.M., et al. (2018). Flooding hazards across Southern China and perspective sustainability measures. Sustainability-Basel, doi: 10.3390/su10051682.
2. Wang, Z.F., et al. (2018). Investigation into geohazards during urbanization process of Xi'an, China. Natural Hazards, doi:10.1007/s11069-018-3280-5.

*Clarified*:    Thanks for recommending the insightful papers, which we have used to strengthen our manuscript. We have added one of them to the reference list, due to the limited number of references (up to 20) allowed in 'Brief Communications'.

---

## Author Response (AR1)

**Point-by-point response to editor's and reviewers' comments**

**Part 1: Author's Responses to Comments from Editor Sven Fuchs**

Comments made by the editor are shown in black text.
Author responses are provided in blue text.

*Comments:* Editor Decision: Publish subject to minor revisions (review by editor) (20 Mar 2019) by Sven Fuchs Comments to the Author:

Dear colleagues, thank you for submitting your manuscript for consideration as a short comment in NHESS. I kindly would like to confirm you the suitability of the topic for the target journal. Moreover, we now received the two comments of the referees as well as your answer to these comments though the open discussion phase.

Based on both I decided that your manuscript needs minor revisions before final acceptance in the "brief communication" section of NHESS. Even if you already uploaded a track-change (comment #3) and clean version (comment #4) of your revised work, I would like to ask you to go once again over the material and check for updates. Please also double-check references in the text body (e.g., Roger Pielke vs. Pielke).

I am looking forward to receiving your finally revised version of nhess-2018-394 as soon as possible. Please do not upload the revised version in the discussion section, but following the link provided in this e-mail.

*Accepted*: Thanks for confirming the suitability of our manuscript and the suggestions for further improvement. We have thoroughly checked our paper, based on the version we published in the Interactive Discussion Section. Specifically, we further improved our figures and double-checked the references.

**Part 2: Author's Responses to Comments from the Anonymous Referee 1**

Comments made by Anonymous Referee 1 are provided in black text.
Author responses are provided in blue text (line and page numbers refer to the clean version).

*General Comments.* This manuscript presents the brief communication of the 1998 flood in China. This topic is related to the scope of this journal. The authors discussed the rapid urbanization and climate change pose new challenges and rethink whether China is prepared for the next mega-flood. However, this manuscript still need to addresses and clarifies several points before it can be accepted. The following comments may help enhancing the quality of this work.

*Accepted*: Thanks for confirming the relevance of our manuscript and the suggestions for

further improvement. We have thoroughly revised our paper, addressing your valuable comments and suggestions.

***Detailed Comment 1***. Scientific writing: The manuscript must be professionally proofread and edited. In addition, the authors may pay attention to some aspect of the conventional research writing. Although it is a brief communication, the structure of the manuscript should be enough, especially the connection between the sentences, the components/structure of the key parts (Abstract, Introduction, body, Conclusion).

*Accepted*:    Thanks for the suggestion. After revising the manuscript, an additional, thorough, proofreading of the manuscript has been carried out by one of the co-authors (Ward), who is a native English speaker and has extensive publication experience in scientific journals. Furthermore, in the revised version we improved the following aspects:

- We rewrote the Abstract to make it more self-explanatory (also following your *Detailed Comment 4*).
- We have improved the connections between sentences and sections to enhance the logical flow. With regards the structure, we checked the journal's guidelines and several recent papers of the type 'Brief Communication', and found that the structure is in accordance with these. This structure is typical for an opinion paper with a 'Brief Communication' type; several similar examples are listed below (Please check the references below).

**Detailed Comment 3**.    The short communication should discussion the persective of flood disaster management, e.g. flood risk assessment and prediction. The following publications are refered: "Flood risk assessment in metro systems of mega-cities using a GIS-based modeling approach" "Assessment of geohazards and preventative countermeasures using AHP incorporated with GIS in Lanzhou"
**Accepted**:    We enhanced the discussion on flood management with the information of the recommended papers, and added one of them as reference. Note the limited number of references (up to 20) allowed in 'Brief Communications'.

**Detailed Comment 4**.    The abstract is too short. Although this is a short communication, the problem, method and results should be included in the abstract.
**Accepted**:    Thanks for this comment. Following your suggestion, we rewrote the Abstract. Note that there is a 100-word maximum limit, which explains the brevity. It now has 100 words and reads as follows:

> "A mega-flood in 1998 caused tremendous losses in China and triggered major policy adjustments in flood-risk management. This paper aims to retrospectively examine these policy adjustments and discuss how China should adapt to newly emerging flood challenges. We show that China suffers annually from floods, despite large-scale investments and policy adjustments. Rapid urbanization and climate change will exacerbate future flood risk in China, with cascading impacts on other countries through global trade networks. Therefore, novel flood-risk management approaches are required, such as a risk-based urban planning and coordinated water governance systems with public participation, in addition to traditional structural protection".

**Part 3: Author's Responses to Comments from the Anonymous Referee 2**

Comments made by Anonymous Referee 2 are shown in black text.
Author responses are provided in blue text (line and page numbers refer to the clean version).

***General comment***:   This paper reviewed a mega-flood in 1998 which caused tremendous losses in mainland China. Since rapid urbanisation and extreme climate result in great challenges, novel flood risk management is in urgent need. The findings of this study seem to have a guiding role for efficient flood risk management, but there're some issues need to be addressed prior to the acceptance of paper publication in NHESS. Additionally, the authors may pay attention to some aspect of the conventional research writing, especially the connection between the sentences, the components/structure of the key parts (Abstract, Introduction, Body, and Conclusions). I suggest the authors may read the following references to modify the paper accordingly. Glasman-Deal, H. (2010). Science Research Writing for non-native speakers of English. Imperial College Press, London, 228p.

*Accepted*:   Thanks for the acknowledgement of the importance of our manuscript and the suggestions for further improvement. We made a thorough revision, performed additional proof reading, and in particular improved the connections between sentences and sections to enhance the logical flow. Note that the chosen structure is typical of the 'Brief Communication' papers in NHESS; several similar examples are listed below, which are also used in other NHESS papers (Please check the references below).

***Clarified***: Thanks for recommending the insightful papers, which we have used to strengthen our manuscript. We have added one of them to the reference list, due to the limited number of references (up to 20) allowed in 'Brief Communications'.

[revised manuscript text omitted]

---

## Author Response (AR2)

**Author's Responses to Comments from Editor Sven Fuchs**

Comments made by the editor are shown in black text.
Author responses are provided in blue text.

***Comments:*** One last point: As indicated earlier, please double check the references; according to the NHESS style it is (a) foreseen that all authors are mentioned in the list (no "et al."), (b) please provide full journal titles and (c) please indicate a doi for all the cited contributions. It is also advisable that you check capitalization of titles carefuly, some are, some not.

***Accepted:*** Thanks for indicating our mistakes with the references, which are mainly because we mistakenly relied on an online Endnote style in preparing the references. We have double checked all the references according to the NHESS style carefully. First, we mentioned all authors in the Reference section. Second, we used the full journal names instead of abbreviations. Third, we included a doi for all the references applicable. In the revised reference list, only two books and a Chinese journal paper did not have a doi, for which doi is not applicable. Fourth, we checked the titles and applied a headline-style capitalization consistently.

Furthermore, we carefully checked our manuscript to make sure it follows the NHESS guidelines.

[revised manuscript text omitted]

5  Mei, J.: Flood Control Planning and Construction in China , China Water Resources, 17-19, 2010 (in Chinese).

MWR: The 1998 China Flood, China Water Power Press, Beijing, 1999.

MWR: China Water Statistical Yearbook 2017, China Water Power Press, Beijing, 2017 (in Chinese).

MWR: Flood Prevention Plan of Yangtze River, Ministry of Water Resources, Beijing, available at: http://www.cjw.gov.cn/xwzx/ztjjx/zjfyhsfa (last access: 6 June 2018), 14 pp, 2015 (in Chinese).

10  Opperman, J. J., Galloway, G. E., Fargione, J., Mount, J. F., Richter, B. D., and Secchi, S.: Sustainable Floodplains through Large-Scale Reconnection to Rivers, Science, 326, 1487-1488, https://doi.org/10.1126/science.1178256, 2009.

Ouyang, Z., Zheng, H., Xiao, Y., Polasky, S., Liu, J., Xu, W., Wang, Q., Zhang, L., Xiao, Y., Rao, E., Jiang, L., Lu, F., Wang, X., Yang, G., Gong, S., Wu, B., Zeng, Y., Yang, W., and Daily, G. C.: Improvements in Ecosystem Services from Investments in Natural Capital, Science, 352, 1455-1459, https://doi.org/10.1126/science.aaf2295, 2016.

15  Pielke, R.: The Precipitous Decline in US Flood Damage as a Percentage of GDP, available at: https://theclimatefix.wordpress.com/2015/02/05/the-precipitous-decline-in-us-flood-damage-as-a-percentage-of-gdp/,  (last access: 10 October 2018), 13 pp, 2013.

Ward, P. J., Jongman, B., Aerts, J. C. J. H., Bates, P. D., Botzen, W. J. W., Diaz Loaiza, A, Hallegatte, S., Kind, J. M., Kwadijk, J., Scussolini, P., and Winsemius, H. C.: A Global Framework for Future Costs and Benefits of River-Flood Protection in Urban Areas, Nature Climate Change, 7, 642-646, https://doi.org/10.1038/nclimate3350, 2017.

20  Willner, S. N., Otto, C., and Levermann, A.: Global Economic Response to River Floods, Nature Climate Change, 8, 594-598, https://doi.org/10.1038/s41558-018-0173-2, 2018.

Winsemius, H. C., Aerts, J. C. J. H., van Beek, L. P. H., Bierkens, M. F. P., Bouwman, A., Jongman, B., Kwadijk, J. C. J., Ligtvoet, W., Lucas, P. L., van Vuuren, D. P., and Ward, P. J.: Global Drivers of Future River Flood Risk, Nature Climate Change, 6, 381-385, https://doi.org/10.1038/nclimate2893, 2016.

25  Zhou, B., Wen, Q. H., Xu, Y., Song, L., and Zhang, X.: Projected Changes in Temperature and Precipitation Extremes in China by the CMIP5 Multimodel Ensembles, Journal of Climate, 27, 6591-6611, https://doi.org/10.1175/jcli-d-13-00761.1, 2014.